# INVESTIGATION OF ROOTZONE SALINITY WITH FIELD MONITORING SYSTEM AT TSUNAMI AFFECTED RICE FEILDS IN MIYAGI, JAPAN

Ieyasu Tokumoto<sup>\*1</sup>, Katsumi Chiba<sup>2</sup>, Masaru Mizoguchi<sup>3</sup>, and Hideki Miyamoto<sup>1</sup> <sup>1</sup>Department of Environmental Sciences, Saga University, Saga, Japan

 <sup>2</sup>Department of Environmental Sciences, Miyagi University, Sendai, Japan
 <sup>3</sup>Department of Global Agricultural Sciences, University of Tokyo, Tokyo, Japan Correspondence to: Ieyasu Tokumoto (<u>vasu@cc.saga-u.ac.jp</u>)

#### ABSTRACT

After the 2011 Tohoku earthquake, thirteen thousand hectares of farmlands were damaged by massive Tsunami near coastal sites in Miyagi, Japan. Some eighty percent of the damaged farmlands have been recovered in 2014, but subsidence and high salinity groundwater make it difficult to completely remove salinity from the soil. To solve the problem, management of saltwater intrusion plays an important role in

- rootzone salinity control with the Field Monitoring System (FMS), which is remote sensing technology of wireless real-time soil data through the internet data sever to investigate high soil moisture and high salinity in tsunami affected fields. Using the FMS with the time domain transmission system, we monitored soil moisture, electrical conductivity (EC), groundwater level, and EC of groundwater at tsunami damaged
- paddy fields. The field measurements of the FMS were conducted at two sites of tsunami damaged farmlands in Iwanuma and Higashimatsushima of Miyagi prefecture, Japan. After the Tohoku disaster, co-seismic subsidence of 17-21 cm and 50-60 cm of the land was reported at the sites, respectively. Our findings were high EC of groundwater (> 35 dS m<sup>-1</sup>) due to intrusion of sea water into groundwater in 2013.
- Although the shallow groundwater provided salinity to the soil surface in 2014, the

FEM allowed us to monitor high EC (

- Some eighty percent of the damaged farmlands have been recovered in the whole damaged areas of Miyagi (Earthquake and Reconstruction Division of Miyagi Prefecture, 2014) (Fig. 2a and 2b), although underdrains, which can remove salinity under saturated soil condition, have been installed at a part of recovered paddy fields (Fig. 2c and 2d). Chiba et al. (2012) investigated the effect of underdrains on desalination process due to rainfall in paddy fields in Sendai Plain with/without underdrains, suggesting that underdrains facilitated the desalination during rainy season between June and September. However, subsidence and high salinity groundwater, making it difficult to completely remove salinity from the soil near the coastal side all
- Field Monitoring System (FMS) to measure soil moisture and salinity developed by Mizoguchi et al. (2012) can facilitate recovery of agricultural production on damaged lands (Fig. 3). The FMS contains three main components: data loggers (Decagon Device and/or Campbell Sci.), soil/meteorological sensors, and a field router (FR)(X-ability), allowing us to collect the in-situ data through Bluetooth and send it to the data server over the Internet. The FR mainly consists of a micro-PC, 3G/GSM USB

over the Miyagi (Chiba et al., 2015).

- modem, webcam, and battery in a waterproof dust-tight box. The FR consumes little power (six-watt solar panel), is easy to set up, and transmits data inexpensively over a cellular mobile line. With the FMS, Chiba et al. (2012) monitored electric conductivity (EC) of groundwater and groundwater level successfully. However, monitoring high
- soil water content ( $\theta$ ) and high bulk EC caused by salt leaching and saltwater intrusion was limited. To monitor high bulk EC in soil, transmission line, electromagnetic measurement methods such as time domain reflectometory (TDR) and time domain transmission (TDT) would be widely accepted. Especially, TDT sensor (Acclima) is more affordable than TDR sensor due to low cost, high precision of  $\theta$  estimation, and

- less user-ability requirement of TDT waveform analysis (Blonquist Jr. et al., 2005). Miyamoto et al. (2013) examined bulk EC affecting permittivity with TDT devices through the obtained TDT waveform analysis in sand, loam, and clay. Their results showed a threshold of 5 dS m<sup>-1</sup> when maximum slope of TDT waveform for bulk EC estimation was too low to detect.
- Understanding the desalinization process and the management of saltwater intrusion in coastal aquifers is a critical challenge in the tsunami damaged farmlands. There have been hydrological studies regarding saltwater intrusion under natural conditions for more than 100 years (Barlow, 2003), but the degree of saltwater intrusion varies widely among localities and hydrogeologic conditions. For the management of 85 saltwater intrusion after subsidence, a new approach is required in the Tohoku earthquake damaged land (Sasaki, 2014; Miyauchi, 2015).

The aim of this study was thus to investigate temporal change in rootzone salinity of the tsunami damaged farmlands in Miyagi using the FEM with TDT sensor system to compare  $\theta$  and high bulk EC with the saline groundwater level. This was achieved at two farmlands, completed clearance of disaster debris due to the MAFF guideline of desalinization process. In addition, this paper presents the current status on a construction method for the saltwater intrusion management along the Pacific coast.

### 95 2. METHODOLOGY

### 2.1 Study site

The field measurements of the FMS were conducted at two sites of tsunami damaged farmlands: Iwanuma (38°03'17.8"N 140°54'50.3"E) and Higashimatsushima

(38°25'38.6"N 141°14'46.4"E) in Miyagi, Japan (Fig. 4). The paddy fields in Iwanuma are located at a costal side, 0.7 km and 0.25 km away from the Pacific Ocean and Abukuma River estuary, respectively (Fig. 4a). After the disaster, co-seismic subsidence of 17-21 cm of the land was reported by Ozawa et al. (2011), and the area was covered by the top of 10 cm thick sediment layer consisting of sand and/or mud (Chiba et al., 2012). The geomorphology of the low-lying coastal land prevented much of the

- seawater from draining away, resulting in water ponding for 20 days. In 2013, drainage trench and canals in the area were recovered, and deposition of debris such as woods and rock fragments was stripped from the soil surface. However, much of the area near Iwanuma has high EC of groundwater due to the low sea level (approximately 0 m). In Higashimatsushima, the tsunami damaged area (3600 ha) consisted of 40% farmlands,
- in combination with subsidence of 60 cm and deposit of 50-cm sea sand layer, resulting in 10-cm lower land prior to the disaster (Ota et al., 2013). In 2014, debris was removed, but underdrain for desalination process was not constructed in the areas.

The regions are characterized by a warm humid temperate climate (mean monthly temperatures ranged from 1.6 °C to 24.2 °C from 1981 to 2010), and average of annual rainfall is approximately 1100 mm (Japan Meteorological Agency, 2015). Monthly rainfall over 150 mm was recorded between July and September, but typhoons are rare in the regions. Light snow (maximum snow accumulation < 10 cm) occurs between December and February. For seasonally rainfall frequency, summer time is the best season to reduce soil salinity. In general, EC of rainfall near the costal side ranges

from 1.7 to 8.58 mS m<sup>-1</sup> (Environment Policy Division of Miyagi Prefecture, 2005), suggesting that chloride deposition from sea spray is low.

2.2 TDT sensor calibration

Calibration of TDT sensor (Acclima) was conducted in a walk-in controlled-temperature laboratory to perform at the standard temperature of 25°C. A

- 9.8-cm i.d.  $\times$  20-cm long cylindrical column was filled with sieved and air-dried soils (dominated by organic-rich and muddy sediments) of Iwanuma and Higashimatsushima above 2-cm sand layer that was at the bottom of the column, and was packed to a dry bulk density of 1.50 g cm<sup>-3</sup>. A TDT sensor was inserted vertically in the center of the column. EC<sub>1:5</sub> (1:5 soil to water extract EC) of tsunami affected soil from Iwanuma and
- Higashimatsushima was 0.75 dS m<sup>-1</sup> and 2.33 dS m<sup>-1</sup>, respectively. For the reference, we used the desalinized soil ( $EC_{1:5} = 0.24 \text{ dS m}^{-1}$ ) of Higashimatsushima that was washed with distilled water. Water was allowed to infiltrate in a stepwise manner from the bottom at a rate of 0.5 to 2 cm d<sup>-1</sup> over a 3-day period to create a range of water contents, which is known as the rapid upward infiltration method proposed by Young et
- al. (1997). The weight of the soil column was monitored with a digital balance to calculate the average of water content in the influence area of TDT sensors. We confirmed that there was no influence of the sand layer on TDT measurements. After the experiment was completed, average volumetric water content in the entire soil column,  $\theta_{ave}$  and bulk density,  $\rho_{b}$  were obtained gravimetrically.
- Permittivity and EC data of the TDT sensor were determined with SDI-12 (digital signal processing algorithms) of CR800 (Campbell Sci.), showing travel time and maximum slope of the TDT waveform. The resulting digitized waveform can be analyzed by taking the first and second derivatives and extracting maximum inflection points and maximum slope points in the waveform (Blonquist Jr. et al., 2005). While
- the entire TDT waveforms were not received by CR800, permittivity and EC were calculated from the readings of maximum inflection points and maximum slope. Before the experiment, bulk EC of TDT sensor was calibrated with salt water.

### 2.3 FMS with TDT sensors

- In the summer of 2014, we began a study at the sites (Iwanuma and Higashimatsushima) to monitor soil moisture and bulk EC using FMS with TDT sensors (Acclima) (Fig. 4b and 4c). Three TDT probes were installed at depths of 10, 20, and 40 cm. TDT measurements were recorded every 1 h by a model CR800 datalogger (Campbell Scientific, Logan, UT) and uploaded at the data server at noon through the FR (X-ability). Groundwater level (G.W.L.) and EC of groundwater were measured
- with a CTD sensor (Decagon Devices), hooked up with a Em50 datalogger (Decagon Devices). The CTD sensor was placed in a well at a depth of 50 cm from the soil surface.

### **RESULTS AND DISCUSSION**

#### **TDT Sensor Calibration**

- The relationship between ε and θ, obtained with salinity (EC<sub>1:5</sub> = 2.33 dS m<sup>-1</sup>) and desalinized soils (EC<sub>1:5</sub> = 0.24 dS m<sup>-1</sup>) from Higashimatsushima, was very close, although θ (ε > 21) of the salinity soil was slightly higher than the desalinized soil (Fig. 5). When saturated θ for the soil was 0.55 m<sup>3</sup> m<sup>-3</sup>, bulk EC of the desalinized soil was 6 dS m<sup>-1</sup>, which caused a considerable measurement error (ε was equal to zero). However, we did not find such errors in the desalinized soil of Higashimatsushima at saturated
- condition. Likewise, no error readings occurred in the salinized soil of Iwanuma (EC<sub>1:5</sub> =  $2.33 \text{ dS m}^{-1}$ ), resulting in a threshold of 6 dS m<sup>-1</sup> for those tsunami affected soils.

Measured  $\varepsilon - \theta$  relationship was nearly identical to Topp Eq. (Topp et al., 1980), but Topp Eq. slightly underestimated  $\theta$  (Fig. 5). For more accurate  $\theta$  estimation, we applied a multipler of -1.15, obtained by a regression analysis to Topp Eq., yielding the equation

 $\theta = 1.15$ 

 $\begin{array}{l} \times \ (-5.3 \times 10^{-2} + 2.9 \times 10^{-2} \varepsilon - 5.5 \times 10^{-4} \varepsilon^2 + 4.3 \\ \times \ 10^{-6} \varepsilon^3 ) \qquad [1] \end{array}$ 

with an  $r^2$  of 0.98 for a value of  $\varepsilon$  in the range from 3.6 to 31.5. On the other hand, the  $\varepsilon$ - $\theta$  relationship for the soil of Iwanuma was fitted well with Topp Eq. ( $r^2 = 0.97$ ) (data is not shown).

# 175 Soil Moisture and EC Monitoring by FMS at Iwanuma

In October 2014, the Iwanuma region was hit by two typhoons, caused rainfall events over 112 mm, while typhoons are rare (Fig. 6a). Soil water content drastically increased at depths of 10, 20, and 40 cm on 5 October and  $\theta$  below 20 cm depth was saturated for 20 days (Fig. 6b). Saturated hydraulic conductivity was 0.35 and 9.76 cm

d<sup>-1</sup> at depths of 15 and 25 cm, respectively. Bulk EC increased with increasing θ, and the highest bulk EC was 3.7 dS m<sup>-1</sup> at a depth of 40 cm (Fig. 6c). Shallow groundwater level also increased up to 20 cm and 40 cm above the soil surface on 5 October and 13 October, respectively (Fig. 6d). An example of webcam photos showed the ponding condition after the heavy rainfall on 5 October (Fig. 7), suggesting that EC of groundwater decreased from 22 dS m<sup>-1</sup> to 5 dS m<sup>-1</sup> due to a direct inflow into a well. For this reason, increase in EC of groundwater was found after 5 October (Fig. 6d). This indicated the intrusion of salt wedge into the groundwater, which has a seasonal cycle of salinity variation due to the strongly seasonal rainfall.

EC of the ocean near the experimental site was 45 dS m<sup>-1</sup>. In 2012, EC of groundwater was 25 dS m<sup>-1</sup> at a depth of 70 cm where the intrusion of salt wedge into the groundwater occurred, while the average of groundwater level was approximately 25 cm depth (Chiba et al., 2015). To decrease the groundwater level, a pumping station was recovered in 2012. The pumping station was used to sustain rotational upland fields

from rice farming to wheat or soybean and maintain the groundwater level (45 cm to 50
cm below surface), which was pumped into water drainage channels that finally drained into the sea. Rice farming was confirmed to remove salt concentration out of the agricultural fields in the region (Chiba et al., 2015), but the high salt concentration (EC > 0.9 dS m<sup>-1</sup>) was resulted in dead soybean (Hoshi and Yusa, 2012). At the end of October 2014, the groundwater level was controlled by the pumping station, resulting in daily variation of the groundwater level from soil surface to 50 cm depth (Fig. 6d). In spring 2015, the pumping station operated simultaneously to maintain the groundwater below a depth of 50 cm (data not shown).

An important question for the Iwanuma site is whether the land subsidence due to the earthquake causes the saltwater intrusion into the groundwater, which is characterized as "transition zone" between freshwater and seawater, containing concentrations of chloride ranging from about 250 to 19,000 mg L<sup>-1</sup>. Based on Miyauchi (2015), who simulated the saltwater movement from Abukuma River estuary with -1 m of sea level of Tokyo Bay, possible contamination of the saltwater (concentrations of chloride = 500 mg L<sup>-1</sup>) occurred even at approximately 200 m away from the sea shore. Our bulk EC data were consistent with the chloride concentration using relationship between chloride concentration and EC proposed by Kaneko et al. (2002):

Chloride concentration (mg 
$$L^{-1}$$
) = 270.86× EC (dS m<sup>-1</sup>) Eq. [2]

# Soil Moisture and EC Monitoring by FMS at Higashimatsushima

During October 2014, total rainfall was 211 mm (Fig. 8a), and  $\theta$  was completely saturated at a depth of 40 cm (Fig. 8b). Bulk EC at a depth of 20 cm exceeded 6 dS m<sup>-1</sup>, which was the maximum detectable value of EC with TDT sensors (Fig. 8c), resulting in error of  $\theta$  measurements at the same depth. This suggests that

coated TDT probes would be required to monitor  $\theta$  in such high salt concentration. To 220 decrease salt concentration near soil surface, daytime pumping was operated (Fig. 8d). A mismatch of daily variations between groundwater level and unsaturated  $\theta$  condition may be explained by low saturated conductivity of the soil (Ks = 1 cm d<sup>-1</sup>). Without exceptions of indirect inflows after typhoons, however, EC of groundwater was about 10 dS m<sup>-1</sup> at a depth of 50 cm.

- Locally, we confirmed the trend of bulk EC decline from 9.4 dS m<sup>-1</sup> to 2.44 dS m<sup>-1</sup> near the soil surface, indicating the effect of rainfall leaching through 2 yr-period, but we found that the subsidence of 60 cm in Higashimatsushima made it difficult to remove salt movement from salty groundwater in freshwater-saltwater environments of the Pacific Coast. Comparison of groundwater level in the Higashimatsushima with
- Iwanuma sites showed that Higashimatsushima site was flooded more frequently, even though the site was covered with tsunami deposit of 50 cm. The profile of EC of groundwater increased linearly as soil depth increased (Fig. 9). As we mentioned, rainfall could maintain the freshwater zone to decline soil salinity near the soil surface. However, it seemed that increase in salty groundwater level caused salination below the
- soil surface. This explained why the shallow groundwater could be salinity source to the soil surface after the subsidence. To solve the salinity problem, raising the ground level of the agricultural fields would be needed (Chiba et al., 2012); nevertheless it represents a costly soil construction method which can also lead to issues of a soil stock from mountain side (Sasaki, 2014).
- Instead of removal of tsunami sediments and underlying soil, Ota et al. (2013) suggested the use of tsunami sediments for agricultural soil because their results showed that tsunami sediments were not contaminated by high concentration of heavy metals such as copper, lead, and arsenic. If the underlying soil is turn over on tsunami

sediments, it would be feasible and sustainable options to maintain fertile soil in the root

zone. However, Ota et al. (2013) also insisted on the need of a construction method to prevent ongoing saltwater intrusion.

### Control of salinity movement from the costal side

For control of saltwater intrusion, the surface water canals and levee system was proposed to convey freshwater from inland water conservation (storage) areas to coastal areas, where the water is recharged through the canals to the underlying aquifer to slow saltwater intrusion in the aquifer (Barlow and Reichard, 2010). In addition to the conventional method, an innovative approach is being proposed to manage saltwater intrusion along the Pacific coast (Ota et al., 2013; Sasaki, 2014), which is the technique to maintain freshwater zone under the tsunami affected agricultural fields. Figure 10 shows the schematic figure of the surface water canal to show the amount of surfacewater canal, Q (m<sup>3</sup> h<sup>-1</sup>) (Sasaki, 2014):

$$Q = \frac{N \cdot H^2 \cdot L}{2d}$$
[3]

where *N* is hydraulic conductivity (m h<sup>-1</sup>), *H* is water level difference between the Ocean and canal, *L* is length of the canal, and *d* is distance between the ocean and canal.

Two dimensional saltwater intrusions with the surface water canal, simulated by Ota et al. (2013), showed that saltwater contamination (concentration of 500 mg L<sup>-1</sup> chloride) can be avoided in root zone at d = 40 m. Without the canal scenario, their simulation result was similar to our observed EC data shown in in Figs. 8c and 9, although our survey was only conducted at a small area of 200-m away from the costal side. This suggests that the recovery of agriculture needs to recharge for the freshwater zone due to water supply near the costal side. To date, no surface water canal has been constructed in the severe subsidence region where the possibility of contributing to control of salt-water intrusion would be a very exciting proposition.

As mentioned in the Introduction, so far some eighty percent of the damaged 270 farmlands have been recovered in terms of removal of tsunami sediments and underlying soil through the guideline by MAFF (2011). However, our results show that local variability of subsidence and high salinity groundwater were affected to remove salinity from the farmlands. As the surface water canal is constructed, it would be necessary to investigate salt concentration with the FMS in terms of assessment of 275 salinity movement from the costal side. The FMS with TDT sensor system was easy to setup and retrieve in-situ data through the Internet.

#### CONCLUSIONS

Our study shows that investigation of rootzone salinity with the FMS at tsunami affected rice fields in Miyagi, Japan. The FEM performed adequately to monitor high

bulk EC (≈ 5 dS m<sup>-1</sup>) even at high θ (≈ 0.54 dS m<sup>-1</sup>) for the effect of rainfall leaching through 3 yr-period after the 2011 disaster. Our finding was decline in bulk EC due to the rainfall leaching, but the shallow saline groundwater made it difficult to remove salinity from the tsunami damaged field in Higashimatsushima where severe subsidence occurred. The EC of groundwater seemed to increase because of saltwater intrusion in groundwater. For the recovery of the damaged fields, the management of saltwater intrusion is also the key components of the recovery plan to maintain the large-scale aquifer storage along the coastal side.

This paper summarized a new approach to manage saltwater intrusion along the Pacific coast with the surface water canal system, proposed by (Ota et al., 2013; 290 Sasaki, 2014). Based on temporal changes in bulk EC profiles and groundwater level, the surface water canal system is required. To continue monitoring and scientific analysis of the entire hydrologic system, therefore, the utilization of FMS is expected to provide positive feedback for the recovery of agricultural production.

# 295 ACKNOWLEDGEMENTS

The research was supported by a grant from the Japan Society for the Promotion of Science (Research Project Number: 26511009).

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
