# Peer review of "INVESTIGATION OF ROOTZONE SALINITY WITH FIELD MONITORING SYSTEM AT TSUNAMI AFFECTED RICE FEILDS IN MIYAGI, JAPAN"

_SOIL, 2016_

## Referee Comment (RC1) · Anonymous Referee #1 · 10 Apr 2016

Work this study has done sounds very important. Especially, temporally precise monitoring of field soil moisture, EC, temperature and G.W.L and transfer those data through ICT must be valuable. In fact, those field conditions vary with rather short time range. Daily or weekly sampling cannot declare what is going in the field. Human behavior, operation of pumping station in this study, may affect soil water regime in the field. Also, visual data collection may be beneficial at field monitoring. In case of curious response of sensors user can check the real situation visually.

It is very interesting that even in the paddy fields those have dense hard pan below surface soil, fluctuation of ground water due to pump operation could affect response of soil sensors. It is interesting what would happen in the upland field in the coastal area.

This text has good contents, however writing, wording, grammar and structure, has many problems. I would think those problems must be fixed before discussion on scientific contents of this paper.

INTRODUCTION section
1. P3, L60-63: It is not clear how FMS can facilitate recovery of damaged lands? Rather, this issue may be one of the aims of this text.
2. P3.L66- "The FR consumes…" might be "The FR consumes less power and can operate with six-watt solar panel." Please check and re-consider the expression.
3. P4, L75-79: The reference of Miyamoto et al. (2013) deals only sand and does not have information on clay and loam soils. Also, as English "There results showed…..too low to detect" seems unclear.
4. P4, L80-85: I would recommend previous papers dealing ground water at Sendai plain, i.e. Uchida et al. (2005) Study on the Subsurface Thermal Structure at the Sendai Plain 1. Construction of 3D regional groundwater flow and heat transport model, J. Geothermal Res. Soc. Japan (27(2), 115-130. General description such as Barlow (2003) sounds weak.
5. P4, L88 and others: Does "FEM" mean specific content or simply miss spell of the "FMS"?
6. P4, L92 I would feel the phrase "…. Current status on a construction method for…." does not make sense. Authors had better write more details and with reader friendly attitude.

   METHODs section
7. P5, L108 and others: "…low sea level" is a description on sea water. It may be "low altitude"
8. P5, L108-111: I cannot understand nor guess what "In Higashimatsushima, the tsunami damaged …….in the areas" means and cannot comment on its contents, neither. Please rewrite it.

9. P5, L117 "Light snow" has alternative mean and text might be "less snow fall (maximum snow depth is less than 10cm) "

10. P5, L119 "… the costal side ranges from …" may be "…coast ranges from …".

11. P7, L151 I could not find figure 4b and 4c.

12. P7? No description on methods relating Figure 9.

RESULTS & DISCUSSION section

13. P7, L160 and others: "salinity soil" may be "saline soil".

14. P7, L163 and others: What does "saturated $\theta$" means? Is this water saturation or it means number of saturated volumetric water content?

15. P7, L163-165: Where we can see data relating "bulk EC of the desalinized soil was 6 dS/m, which caused a considerable measurement error."?

16. P8, L176 and latter part and Figures 6&8: Those figures employ "DOY" system however in the text author uses common day-month description and cause confusion by potential readers, i.e. it is difficult for readers to find which rainfall event corresponds to the typhoon event. I would recommend to use day-month system for both figures and text.

17. P8, L177 I would suggest to delete ", ".

18. P8, L185: "…a direct inflow into a well" may be "…direct inflow of surface water into a well…", isn't it?

19. Here, drop of both EC and G.W.L happened simultaneously at first typhoon, (DOY=278), but for the second typhoon (DOY=288) both EC and G.W.L. rose in a short period. This suggests different behavior of surface and subsurface water in the field but authors did mention on only first typhoon. Author had better descript on the second typhoon.

20. P8, L189 "EC of ocean" may be "EC of sea water", isn't it?

21. P8, L192 "decrease the ground water level" may be "depress the ground water level".

22. P8, L192 and others: Please pay attention on use of "deep" and "in depth".

23. P8, L191-P9, L196 "The pumping station ….into the sea": These sentences are not clear as well it is not precise. Suppose, in Japan, this pumping system operates to transfer surface water to the sea to depress shallow ground water level. However, irrigated fields under arid climate, i.e. central Asia like Uzbekistan, direct pumping of ground water to depress ground water level is quite common. Those are really different and to prevent miss-understanding author had better descript precisely.

24. P9, L196 and others: "remove salt  ….", concentration cannot be removed.

25. P9, L198: "dead soybean" may be "withering of soybean".

26. P9, L198-L200: I would guess "At the end of …..(Fig.6d)." might be "From end of October 2014 and later, ground water level was controlled by the pumping station, constrained daily fluctuation of the ground water level within between soil surface to 50cm in depth".

27. Please consider the descriptions.

28. P9, L201 "pumping station operated simultaneously to maintain the ground water below a depth of 50cm": What "simultaneously" means in this sentence?

29. P9, L2120-213: Eq(2) from Kaneko et al. (2002) employed EC of water for the calculation. In this study, authors showed only bulk soil EC and did not discuss soil water EC (ECw in general). So, it is impossible, at lease in this version, to say bulk EC data were consistent with the chloride concentration.

30. P9, L215 " $\theta$ " may be better to change to "soil"

31. P9, L215-P10,L224: In Fig.8. monitored field in Higashimatsushima was often submerged, positive ground water level". It is very curious that even under surface water ponding monitored water content at 10 and 20 cm in depth showed temporal changes. Why those could change under ponding condition? Also, I would like to know surface condition of the field during the monitored period.

32. P10,L225-228: It sounds that shallow salty ground water prevent (or retardate) desalinization by rainfall (also mentioned at 1st paragraph of CONCLUSION). However, bulk EC is not good to discuss salinization and desalinization since it is affected moisture condition. In Higashimatsusima.at 40cm in depth, soil bulk EC decreased with almost constant soil moisture content. This might be a result of leaching by rain water, the reviewer feels. Oppositely, in Iwanuma where relatively deeper ground water, temporal changes in $\theta$ and bulk EC coincided temporally. Depth of 20cm that showed small EC depression with the constant water content may be a sign of leaching. Overall, author had better descript monitored result carefully and precisely, as possible.

33. P10, L257: What does "made it difficult to remove salt movement from salty ground water" means?

34. P10,L257 Does "fresh water – salt water environment" mean "brackish water environment"?

35. P10, L234: "increase in salty ground water" may be "rise in salty ground water"

36. P10, L234: "salination" may be "salinization"

37. P10, L234-235: It is curious that why "salinization" caused only below the surface soil?

38. In such situation surface soil may also salinized.

39. P10, L240 and others: In the phrase of "tsunami sediments and underlying soil" what under lying soil means? Does this mean remove both "tsunami sediments and underlying soil"? Description is too simple and difficult to be understood by potential readers.

40. P11,L249: Does "inland water conservation" mean "inland fresh water reservoir"?

41. P11,L246-L259: Eq(3) may be an expression of simple horizontal percolation above impermeable bed by Darcy law caused by difference in water level between two points. In basic this equation may express horizontal flow above the water level of drainage canal. For engineering purpose this might be applied in the field however as scientific paper author should mention to or modify the equation.

42. P11, L261 "mg L-1 chloride" may be "mg- chloride L-1"

43. P11, L263: As depicted before on Eq(2), please specify bulk soil EC and soil water EC and revise the description.

CONCLSION

44. "Conlusion" may be thoroughly rivesed after revision on "aim" and "Results and Discussion".

---

## Referee Comment (RC2) · Anonymous Referee #2 · 20 Apr 2016

This paper aims to analyze root zone salinity in rice fields hit by the 2011 earthquake and tsunami in Japan. Flooding with seawater introduced salinity to the soil, and the subsidence associated with the earthquake enhanced sea water intrusion. In order to assess and control salinity, a monitoring system was installed at two sites, and time series of soil water content, ground water level, bulk electrical conductivity and ground water electrical conductivity were obtained. These two datasets are interpreted, and it was found that rainfall reduced salinity but that salinity was reintroduced by high groundwater levels. This suggests that improved shallow aquifer management is required.

After reading this manuscript, I think that it is currently not of sufficient quality to allow

publication in Soil. The manuscript lacks a meaningful scientific focus. In my opinion, a discussion of the monitoring alone is not sufficient. The data need to be interpreted within the framework of a scientific question. In my opinion, the authors have not succeeded in doing this. It remained unclear to me how the measurements help to understand the desalinization process. In addition, the quality of writing is not sufficient in many places, which makes it difficult to understand what the authors are trying to say. It may be appropriate to seek help from a native speaker or an editing service to improve the quality of the writing. Therefore, I cannot currently recommend the manuscript for publication, and I suggest to release it to the authors. Please find below some specific comments for your consideration.

SPECIFIC COMMENTS Line 60-65. The technical details should be presented in the Materials and Methods section. Here, it is sufficient to describe the general need for field monitoring (and which parameters are relevant).

Line 85. A more in-depth discussion of the problem of land subsidence and the consequences for saline water intrusion early in the manuscript would help the reader to better understand the issues at hand.

Line 112. Underdrains were not installed at both sides? Would it not be better to also investigate a site with underdrains to see whether this leads to sufficient desalinization? In any case, it should be made clear why these two sites were selected for the field investigations.

Line 179. How was the saturated hydraulic conductivity determined?

Line 187. It is not clear to me what you mean with "intrusion of salt wedge into groundwater".

Line 189 – 213. Here a lot of statements are made that are not supported by the data directly. It is difficult to see how all the different desalinization strategies add up, and what conclusions can be drawn from the presented data.

Line 248 – 276. This part is not well related to the rest of the paper. It would be nice if future measurements could validate the concept presented here.

---

## Author Comment (AC1) · 4 Jul 2016

**Response to the reviewer1 comments**

Thank you for reviewer comments. They helped strengthen our paper. We revised the manuscript as much as possible in line with the suggestions made by the reviewer 1.

*Reviewer 1*.

**General comments**

Our original manuscript focused on monitoring soil moisture content ($\theta$) and bulk soil EC ($EC_b$) using the Field Monitoring System (FMS) to work on desalinization process of tsunami affected agricultural fields. However, Reviewer 1 noted it lacked a scientific focus. In the revised manuscript, we described what the advantage of FMS is and how meaningful the FMS data is for new approaches: $EC_b$-$\theta$-soil water EC($EC_w$) relationship and two dimensional water and chloride transport for better future scenario of the surface water canal system. Based on field data ($EC_b$, $\theta$, and estimated $EC_w$) and the simulation results, we discussed the saltwater intrusion management of the future surface canal system.

**Specific comments**

1) Line 60-65. The technical details of FMS were shorten in the Materials and Methods section.

2) Line 85. In the introduction section, the problems of land subsidence and the consequences for saline water intrusion were mentioned.

3) Line 112. Also, the current situation of desalinization regarding underdrains was described to explain why we cannot use underdrains at our research sites.

4) Line 179. In the materials and methods section, "soil core sampling method" was added to measure the saturated hydraulic conductivity.

5) Line 187. The sentence was deleted, but the similar content was rewritten as "a rise in saline groundwater…" in results and discussion section.

6) Line 189 – 213. The sentences were rewritten based on the presented data.

7) Line 248 – 276. Previous studies of the future canal system were presented in the introduction. To predict chloride transport using the water surface canal system, we obtained $EC_b$-$\theta$-$EC_w$ relationship and simulation of two dimensional water and chloride transport was carried out. It would be helpful because it was based on in-situ data using the FMS with TDT measurements.